# Core-shell nanoscale coordination polymers combine chemotherapy and photodynamic therapy to potentiate checkpoint blockade cancer immunotherapy

Chunbai He[1], Xiaopin Duan[1], Nining Guo[1,2], Christina Chan[1], Christopher Poon[1], Ralph R. Weichselbaum[2] & Wenbin Lin[1]

Advanced colorectal cancer is one of the deadliest cancers, with a 5-year survival rate of only 12% for patients with the metastatic disease. Checkpoint inhibitors, such as the antibodies inhibiting the PD-1/PD-L1 axis, are among the most promising immunotherapies for patients with advanced colon cancer, but their durable response rate remains low. We herein report the use of immunogenic nanoparticles to augment the antitumour efficacy of PD-L1 antibody-mediated cancer immunotherapy. Nanoscale coordination polymer (NCP) core-shell nanoparticles carry oxaliplatin in the core and the photosensitizer pyropheophorbide-lipid conjugate (pyrolipid) in the shell (NCP@pyrolipid) for effective chemotherapy and photodynamic therapy (PDT). Synergy between oxaliplatin and pyrolipid-induced PDT kills tumour cells and provokes an immune response, resulting in calreticulin exposure on the cell surface, antitumour vaccination and an abscopal effect. When combined with anti-PD-L1 therapy, NCP@pyrolipid mediates regression of both light-irradiated primary tumours and non-irradiated distant tumours by inducing a strong tumour-specific immune response.

[1] Department of Chemistry, The University of Chicago, 929 E 57th Street, Chicago, Illinois 60637, USA. [2] Department of Radiation and Cellular Oncology and The Ludwig Center for Metastasis Research, The University of Chicago, 5758 S Maryland Avenue, Chicago, Illinois 60637, USA. Correspondence and requests for materials should be addressed to W.L. (email: wenbinlin@uchicago.edu).

Approximately 150,000 patients are diagnosed with colorectal cancer in the United States annually, with one-third dying from metastasis[1]. Although the 5-year survival rate for localized colorectal cancer is ∼89%, this number drops to only ∼12% for cancers that have metastasized to the liver, lungs or peritoneum[2].

Stimulation of the host immune system has been shown to generate an antitumour immune response capable of controlling metastatic tumour growth[3–6]. Immune checkpoint blockade therapy, which targets regulatory pathways in T cells to enhance antitumour immune response, has witnessed significant clinical advances and provided a new strategy to combat cancer[7]. Among them, the PD-1/PD-L1 pathway inhibits immune activation by suppressing effector T-cell function[8,9] and is upregulated in many tumours to cause apoptosis of tumour-specific cytotoxic T-lymphocytes and transmit an anti-apoptotic signal to tumour cells[10,11]. Antibody-mediated specific blockade of the PD-1/PD-L1 axis can generate potent antitumour activity in murine tumour models[12,13]. With the exception of metastatic melanoma, the durable responses generated by checkpoint blockade therapy are still low. Although blockade of PD-1 was shown not to be effective in metastatic colon cancer, a recent report by Le et al.[14] demonstrated that PD-1 blockade was effective in a subset of colon cancer patients who were deficient in mismatch repair, reopening the door to immune modulation with interventions such as chemotherapy and radiotherapy to increase the durable response rate[15]. We hypothesize that combining PD-L1 blockade with multimodality nanoscale coordination polymer (NCP) nanoparticles can increase the response rate of checkpoint blockade cancer immunotherapy and perhaps broaden the use of immunotherapy in metastatic colon cancer.

As a new class of self-assembled hybrid nanomaterials composed of metal connecting points and organic bridging ligands[16,17], NCPs have highly tunable compositions and structures, can combine multiple therapeutic agents or modalities[18] and are intrinsically biodegradable. By combining non-toxic photosensitizers, light and oxygen to produce cytotoxic reactive oxygen species, in particular singlet oxygen ($^1O_2$), photodynamic therapy (PDT) kills cancer cells by apoptosis and necrosis, stimulates the host immune system and causes acute inflammation and leukocyte infiltration to the tumours, which increases the presentation of tumour-derived antigens to T cells[19–25]. Oxaliplatin was shown to induce immunogenic cell death (ICD) in murine colorectal cancer models[26].

We herein report the design of NCP nanoparticles that carry oxaliplatin and the photosensitizer pyrolipid (NCP@pyrolipid), to significantly enhance antitumour immunity. NCP@pyrolipid combines two therapeutic modalities, chemotherapy and PDT, to elicit antitumour immunity[27–29], as evidenced by early calreticulin (CRT) exposure on the cell surface, antitumour vaccination, tumour-specific T-cell response and an abscopal effect. The abscopal effect is usually described with ionizing radiation and refers to regression of tumour outside of the irradiated volume. Although the mechanism is unknown, it is thought to be immune modulated. More importantly, NCP@pyrolipid PDT treatment in combination with PD-L1 checkpoint blockade therapy not only led to the regression of the primary tumours, treated locally with light irradiation, but also resulted in the regression of the distant tumours in bilateral syngeneic mouse tumour models of CT26 and MC38 by generating systemic tumour-specific T-cell response with the infiltration of CD8$^+$ T cells and CD4$^+$ T cells in distant tumours.

## Results

**Self-assembly and characterization of NCP@pyrolipid.** Fig. 1 illustrates our overall treatment strategy with multimodality NCP@pyrolipid nanoparticles. NCP@pyrolipid is a core-shell nanostructure with NCP carrying oxaliplatin as its solid core and a self-assembled asymmetric lipid bilayer as its shell (Fig. 2a). The NCP cores were constructed from the coordination polymerization between Zn$^{2+}$ and phosphate groups of the

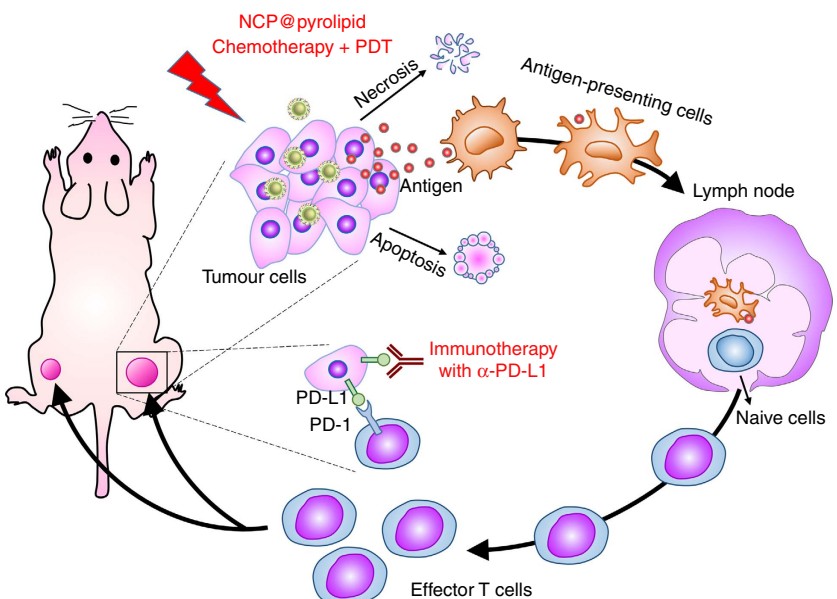

**Figure 1 | Chemotherapy and PDT of NCP@pyrolipid potentiate PD-L1 blockade to induce systemic antitumour immunity.** Chemotherapy and PDT of NCP@pyrolipid induce ICD and an inflammatory environment at the primary tumour site, leading to the release of tumour-associated antigens (TAAs). TAAs are processed and presented by infiltrated antigen-presenting cells, to elicit the proliferation of tumour-specific effector T cells in lymphoid organs, such as tumour-draining lymph nodes. Combined with PD-L1 checkpoint blockade, the NCP@pyrolipid chemotherapy/PDT significantly promoted the generation of tumour-specific effector T cells and enhanced their infiltration in both primary and distant tumours, resulting in not only tumour eradication in the primary sites but also a systemic antitumour immune response to reject distant tumours.

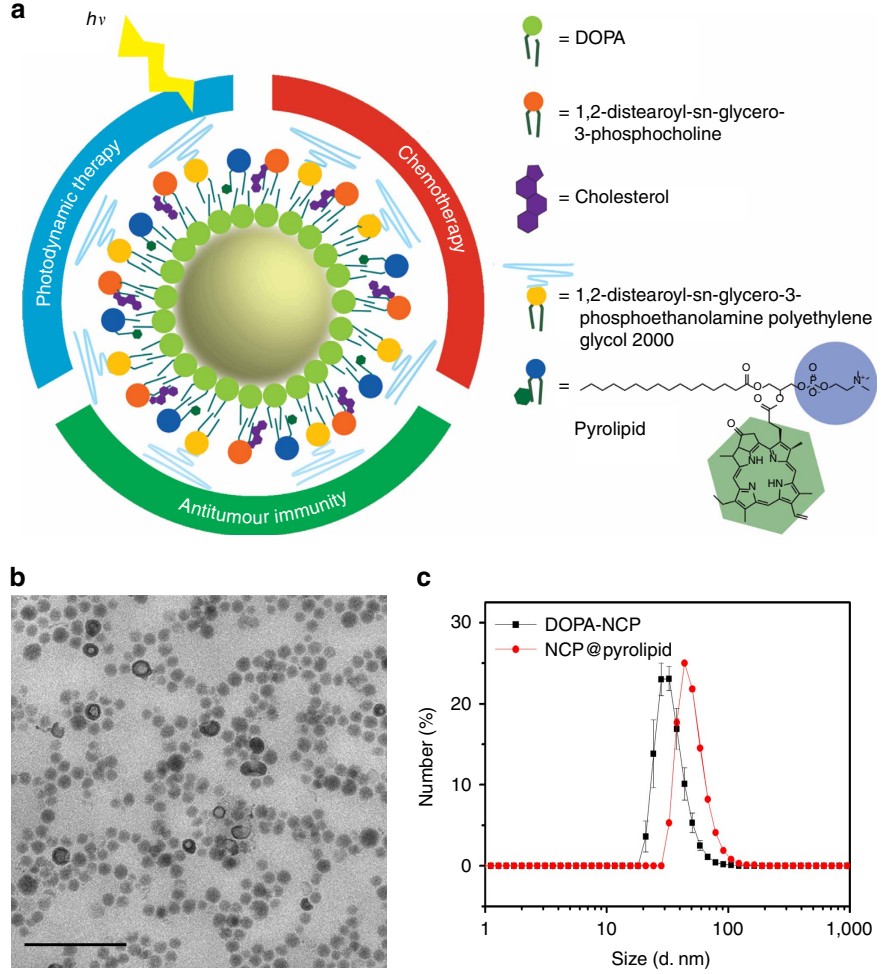

**Figure 2 | Preparation and characterization of NCP@pyrolipid. (a)** Schematic presentation showing the structure of NCP@pyrolipid and its three combined therapeutic modalities. **(b)** Transmission electron microscopy image showing the spherical and monodispersed morphology of NCP@pyrolipid. Scale bar, 200 nm. **(c)** Number-average diameters of DOPA-NCP in THF and NCP@pyrolipid in PBS by DLS measurements. Data are expressed as means ± s.d. (n = 3).

oxaliplatin prodrug, which were further capped with a monolayer of 1,2-dioleoyl-sn-glycero-3-phosphate (DOPA) via Zn–phosphate interactions between NCPs and DOPA molecules, and hydrophobic–hydrophobic interactions among DOPA molecules[16]. Pyrolipid, a lipid conjugate of pyropheophorbide-a, was synthesized via esterification between 1-palmitoyl-2-hydroxy-sn-glycero-3-phosphocholine and pyropheophorbide-a, and incorporated into the outer lipid layer via hydrophobic/hydrophobic interactions. The lipid shell also contains 20 mol% of polyethylene glycol to minimize mononuclear phagocyte system (MPS) uptake and prolong blood circulation after systemic injection.

The spherical DOPA-coated NCP particles possess a Z-average diameter of 55.3 ± 0.2 nm, as determined by dynamic light scattering (DLS; Fig. 2c), and an oxaliplatin loading of 27.6 wt%, as determined by inductively coupled plasma-mass spectrometry (ICP-MS). Transmission electron microscopy image of NCP@pyrolipid demonstrated the formation of uniformly spherical nanoparticles (Fig. 2b). DLS measurements gave a Z-average diameter, number-average diameter, polydispersity index and zeta potential of 83.0 ± 1.0, 51.2 ± 0.1 nm, 0.14 ± 0.01 and − 3.7 ± 0.9 mV, respectively (Fig. 2c and Supplementary Fig. 1), of NCP@pyrolipid dispersed in phosphate-buffered saline (PBS). With intact lipid

bilayer structures, the pyrolipid excited states of NCP@pyrolipid are highly quenched, thereby preventing energy transfer to triplet oxygen, as evidenced by the low amount of $^1O_2$ detected by the singlet oxygen sensor green reagent (Supplementary Figs 2 and 3, and Supplementary Methods). After adding Triton X-100 to NCP@pyrolipid and porphysome (as a control particle)[30–33] to disrupt the lipid bilayer, pyrolipid regained its fluorescence and the released species from both systems efficiently generated similar amounts of $^1O_2$ upon photoexcitation.

The release profiles of oxaliplatin from DOPA-NCP and NCP@pyrolipid were studied in PBS. DOPA-NCP exhibited rapid burst release, with 76% cumulative release of oxaliplatin within 2 h. In contrast, the release of oxaliplatin from NCP@pyrolipid was slow and sustained, with only 5.6% and 21% oxaliplatin released within 2 and 96 h, respectively (Supplementary Fig. 4), suggesting that the lipid bilayer can effectively prevent premature drug release during particle circulation in the blood.

**Investigation of ICD.** CRT is a distinct biomarker exposed on the surface of cells undergoing ICD[34,35]. We first evaluated time-dependent cellular uptake of NCP@pyrolipid from 1 to 24 h in CT26 cells (Supplementary Fig. 5). After the demonstration of efficient cellular uptake and negligible efflux of NCP@pyrolipid in

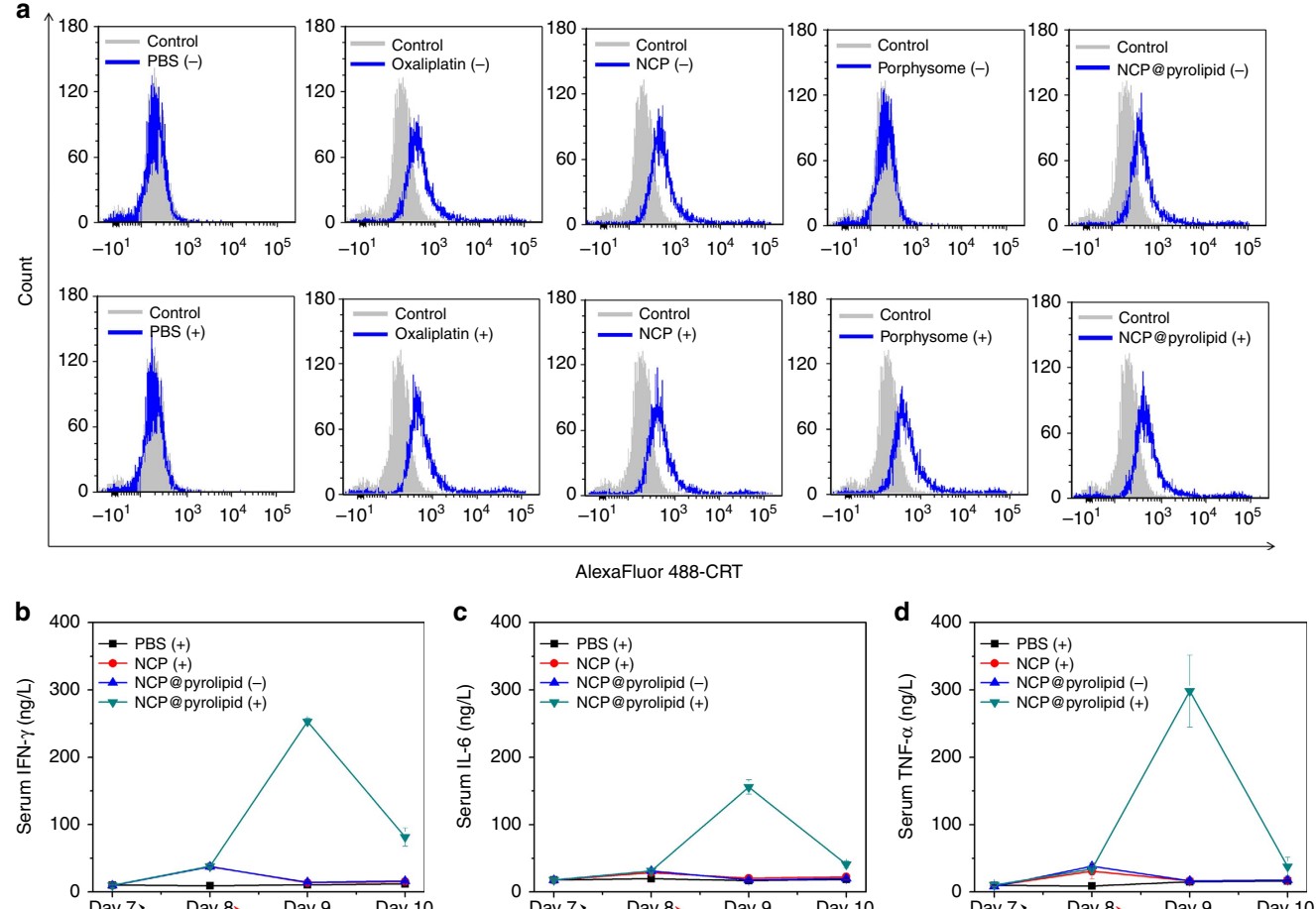

**Figure 3 | NCP@pyrolipid induces ICD and acute inflammation. (a)** CRT exposure on the cell surface of CT26 cells was assessed after the treatments of PBS, free oxaliplatin, NCP, porphysome and NCP@pyrolipid with or without light irradiation (90 J cm$^{-2}$) by flow cytometry analysis. The fluorescence intensity was gated on PI-negative cells. ' + ' and ' − ' in the figure legends refer to treatments with and without irradiation, respectively. **(b–d)** Pro-inflammatory cytokine levels in the serum of mice treated with PDT of NCP@pyrolipid. Syngeneic CT26 tumour-bearing mice were i.v. injected with PBS, NCP or NCP@pyrolipid at an oxaliplatin dose of 2 mg kg$^{-1}$, followed by light irradiation at a dose of 180 J cm$^{-2}$ (670 nm, 100 mW cm$^{-2}$). The blood was collected daily from Day 7, when the mice received their first i.p. injections of nanoparticles, to Day 10, 2 days after the first light irradiation treatment. The serum was separated and the concentrations of IFN-γ (**b**), IL-6 (**c**) and TNF-α (**d**) were determined by enzyme-linked immunosorbent assay. Data are expressed as means ± s.d. ($n = 3$).

CT26 cells, the CRT expression on cells treated with NCP@pyrolipid was determined by flow cytometry (Fig. 3a) and immunofluorescence (Supplementary Fig. 6) and compared with PBS-, oxaliplatin-, NCP- and porphysome-treated cells. We collected and stained cells with Alexa Fluor 488-CRT antibody and propidium iodide (PI) for flow cytometry analysis, where the fluorescence intensity of CRT-stained cells was gated on PI-negative cells. For immunostaining analysis, the cells were stained with Alexa Fluor 488-CRT and 4,6-diamidino-2-phenylindole (DAPI) and observed under confocal laser scanning microscopy (CLSM). Significant amounts of CRT were detected on the surfaces of cells treated with oxaliplatin, NCP or NCP@pyrolipid, regardless of light exposure due to oxaliplatin's ability to induce ICD. Porphysome only induced CRT expression upon light irradiation, suggesting that PDT, not pyrolipid, induces ICD.

We performed an antitumour vaccination experiment to confirm that PDT induces ICD in NCP@pyrolipid-treated cells *in vivo*. We applied PDT treatment to CT26 cells incubated with NCP@pyrolipid *in vitro* to induce ICD, which served as a tumour vaccine when inoculated into BALB/c mice. As shown in Supplementary Fig. 7, mice receiving the NCP@pyrolipid-treated and light-irradiated CT26 cells were protected against a subsequent challenge with live CT26 cells, remaining tumour free in contrast to mice in the control group, which all developed tumours when challenged. This result indicated that PDT of NCP@pyrolipid induced strong ICD in CT26 cells, which acted as an effective vaccine against live tumour cells in immunocompetent mice.

**In vivo antitumour immunity of PDT of NCP@pyrolipid.** To evaluate the antitumour immunity evoked by PDT of NCP@pyrolipid, we collected blood daily from syngeneic CT26 tumour-bearing mice, starting when the mice received their first NCP@pyrolipid injections (Day 7 after tumour inoculation) to Day 10. The serum was separated and analysed by enzyme-linked immunosorbent assay, to determine cytokine production of tumour necrosis factor-α (TNF-α), interleukin-6 (IL-6) and interferon-γ (IFN-γ). Release of such cytokines indicates acute inflammation, an important mechanism in inducing antitumour immunity by PDT[36]. No significant difference was observed in the three pro-inflammatory cytokine levels among control and monotherapy groups during the testing period. However, significantly higher TNF-α (t-test, $P = 7.9 \times 10^{-4}$

versus control), IL-6 ($t$-test, $P = 2.4 \times 10^{-5}$ versus control) and IFN-$\gamma$ ($t$-test, $P = 3.89 \times 10^{-7}$ versus control) levels were observed in mice treated by NCP@pyrolipid with irradiation on Day 9, suggesting that PDT can successfully activate the innate immune response and cause inflammation (Fig. 3b–d). However, 2 days after PDT treatment, all three pro-inflammatory cytokine levels rapidly dropped to their baseline levels, suggesting that the inflammation was only an acute response (Fig. 3b–d).

**Direct cell killing by cytotoxicity.** The cytotoxicity of NCP@pyrolipid was evaluated against two colorectal cancer cell types, murine colorectal adenocarcinoma CT26 and human colorectal adenocarcinoma HT29 cells. By combining the modalities of chemotherapy and PDT into a single nanoparticle, NCP@pyrolipid was expected to both induce apoptosis/necrosis and elicit ICD upon light-emitting diode (LED) light irradiation. As shown in Table 1 and Supplementary Figs 8–11, there was no significant difference in the oxaliplatin $IC_{50}$ of free oxaliplatin, NCP and NCP@pyrolipid in the dark, suggesting that pyrolipid itself does not cause cytotoxicity. However, on irradiation at 54 J cm$^{-2}$ light dose (670 nm), the oxaliplatin $IC_{50}$ of NCP@pyrolipid decreased by ~4- and ~5-fold in CT26 and HT29 cells, respectively. Pyrolipid $IC_{50}$ values also dropped accordingly for NCP@pyrolipid with irradiation, whereas no toxicity was observed for porphysome under either light or dark conditions in either cell line within the same concentration range.

Direct cell killing resulting in apoptosis/necrosis by NCP@pyrolipid with or without light irradiation was evaluated by flow cytometry of cells stained with an Alexa Fluor 488 Annexin V/dead cell apoptosis kit. As shown in Supplementary Table 1, free oxaliplatin, NCP with or without light and NCP@pyrolipid in darkness induced similarly moderate amounts

of apoptosis, 21–26% in CT26 and 17–28% in HT29. Irradiation with light significantly increased the amount of NCP@pyrolipid-treated cells that underwent apoptosis—from 24% and 17% in darkness to 35% and 43% under light in CT26 and HT29 cells, respectively—as well as inducing necrosis in 13.8% and 18.7% of cells, respectively. Cells treated with PBS or porphysome did not show any necrosis or apoptosis, further substantiating our belief that pyrolipid by itself is non-toxic.

***In vivo* pharmacokinetic and biodistribution studies.** A pharmacokinetic and biodistribution study of NCP@pyrolipid by intravenous (i.v.) injection was carried out on CT26 tumour-bearing BALB/c mice (Fig. 4). The distribution of oxaliplatin was quantified by ICP-MS and the concentration of pyrolipid in the blood was quantified by ultraviolet–visible spectroscopy after extraction by methanol as previously reported[18]. The concentrations of both oxaliplatin and pyrolipid in blood over time were fitted by a one-compartment model (Fig. 4b,c). The blood circulation half-lives were determined to be 11.8 ± 1.9 and 8.4 ± 2.6 h for oxaliplatin and pyrolipid, respectively. The difference in their blood circulation half-lives was statistically insignificant ($t$-test, $P = 0.11$). In addition to the long blood circulation time, NCP@pyrolipid exhibited low uptake by the MPS as evidenced by the low % ID g$^{-1}$ (percent injected dose per gram tissue) in liver ($< 7.1 ± 2.5$), spleen ($< 10.4 ± 4.3$) and kidney ($< 9.1 ± 2.5$; Fig. 4a). The peak tumour uptake reached 6.8 ± 1.7% ID g$^{-1}$ 24 h post i.v. injection (Fig. 4a). Intraperitoneal (i.p.) injection increased the tumour uptake of Pt to 10.4 ± 0.7% ID g$^{-1}$ 24 h post injection.

**Anticancer activity in colorectal adenocarcinoma mouse models.** Two colorectal adenocarcinoma mouse models were employed to

**Table 1 | Oxaliplatin and pyrolipid $IC_{50}$ values ($\mu$M) in CT26 and HT29 cells treated with various formulations.**

| | Irradiation* | NCP@pyrolipid | NCP | Oxaliplatin | Porphysome† |
|---|---|---|---|---|---|
| CT26 | √ | 1.00 ± 0.30 (0.22 ± 0.09) | 5.07 ± 1.02 | 4.97 ± 0.49 | > 2.21 |
| | × | 3.97 ± 0.60 (0.88 ± 0.21)‡ | 4.74 ± 0.67 | 5.05 ± 0.95 | NA |
| HT29 | √ | 0.32 ± 0.15 (0.09 ± 0.04) | 1.96 ± 0.47 | 1.87 ± 0.31 | > 2.83 |
| | × | 1.27 ± 0.44 (0.36 ± 0.12)‡ | 1.42 ± 0.49 | 1.44 ± 0.32 | NA |

LED, light-emitting diode; NCP, nanoscale coordination polymer; NCP@pyrolipid, NCP nanoparticles that carry oxaliplatin and the photosensitizer pyrolipid.
The numbers in parentheses refer to pyrolipid concentrations. Data are expressed as means ± s.d. ($n = 6$).
*Cells were irradiated with LED light (670 nm) at 60 mW cm$^{-2}$ for 15 min (equals to 54 J cm$^{-2}$).
†Porphysome containing no oxaliplatin served as control. The amount of pyrolipid in the porphysome was the same as NCP@pyrolipid under the studied concentrations.
‡The dark cytotoxicity comes entirely from the action of oxaliplatin in these formulations.

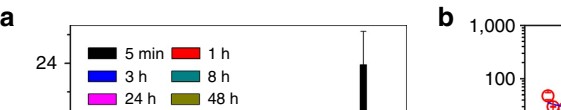

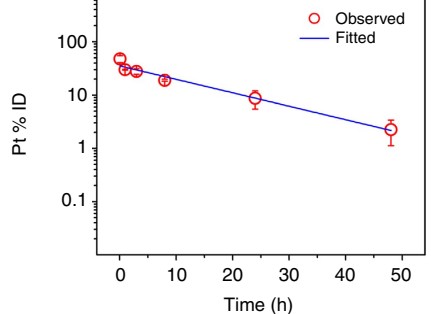

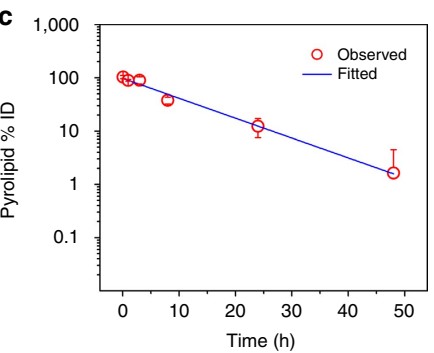

**Figure 4 | Pharmacokinetics and biodistribution of NCP@pyrolipid.** CT26 tumour-bearing BALB/c mice received i.v. injections of NCP@pyrolipid at an oxaliplatin dose of 3 mg kg$^{-1}$ (or pyrolipid dose of 2.1 mg kg$^{-1}$). Pt and pyrolipid concentrations were measured by ICP-MS and ultraviolet–visible spectroscopy, respectively. (**a**) Biodistribution of Pt in mice. A one-compartment model was used to fit the blood concentration of Pt (**b**) and pyrolipid (**c**) over time. Data are expressed as means ± s.d. ($n = 3$).

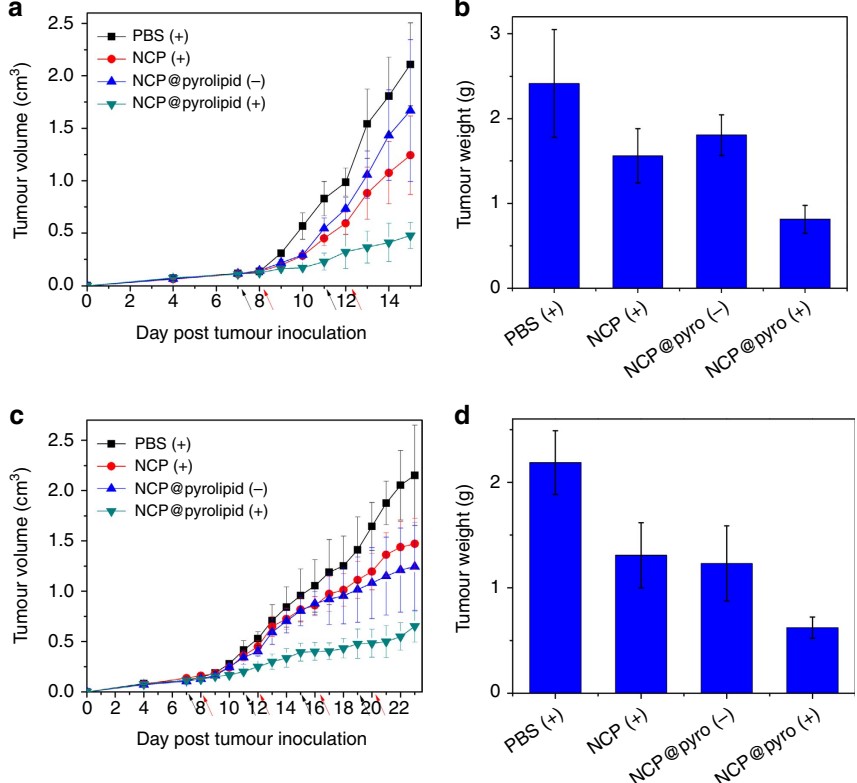

**Figure 5 | *In vivo* antitumour activity of NCP@pyrolipid.** PBS, NCP or NCP@pyrolipid was i.v. injected into a syngeneic CT26 mouse model and an HT29 xenograft mouse model at an oxaliplatin dose of $2\,mg\,kg^{-1}$, followed by irradiation (670 nm, $100\,mW\,cm^{-2}$) for 30 min, 24 h after each injection. Tumour growth inhibition curves in CT26 (**a**) and HT29 (**c**) models. Weights of excised tumours at the endpoint of the experiment for CT26 (**b**) and HT29 (**d**) models. Black and red arrows in **a** and **c** represent the time of drug administration and irradiation, respectively. ' $+$ ' and ' $-$ ' in the figure legends refer to treatments with and without irradiation, respectively. Data are expressed as means ± s.d. ($n=6$).

assess the *in vivo* anticancer activity of NCP@pyrolipid: BALB/c mice bearing murine colorectal cancer CT26 and athymic nude mice with subcutaneous xenografts of human colorectal cancer HT29. Tumour-bearing mice were treated with i.v. injections of (1) PBS, (2) NCP or NCP@pyrolipid (3) in darkness or (4) with light irradiation at equivalent oxaliplatin ($2\,mg\,kg^{-1}$) and pyrolipid ($1.4\,mg\,kg^{-1}$) doses, where applicable. Mice were treated once every 4 days, for a total of two treatments for the CT26 model and four treatments for the HT29 model. Twenty-four hours post injection, the mice in groups (1)–(3) were anaesthetized with 2% (v/v) isoflurane and their tumours were irradiated with a 670 nm LED at an irradiance of $100\,mW\,cm^{-2}$ for 30 min. As shown in Fig. 5a,c and Supplementary Figs 12 and 13, NCP@pyrolipid combined with light irradiation effectively inhibited tumour growth in both CT26 and HT29 models. Without irradiation, NCP@pyrolipid treatment was similar to NCP with irradiation, showing only moderate anticancer efficacy. In combination, these results suggested that neither monotherapy was capable of inhibiting tumour growth, and that the tumour inhibition induced by NCP@pyrolipid was triggered by light activation. PBS control mice possessed CT26 and HT29 tumours weighing ~3.0-fold (*t*-test, $P=7.5\times10^{-6}$) and ~3.5-fold (*t*-test, $P=5.6\times10^{-4}$) more than those treated with PDT of NCP@pyrolipid, respectively (Fig. 5b,d). The resected CT26 and HT29 tumours were subjected to histopathologic analysis, to determine the percentage of cells that underwent apoptosis and necrosis. In both CT26 and HT29 models, mice receiving PBS, NCP or NCP@pyrolipid in darkness had tumours with large areas of viable cancer cells (Supplementary Fig. 14). Tumours from mice receiving NCP@pyrolipid with irradition, however, showed significant amounts of apoptotic cells,

as confirmed by TUNEL (TdT-mediated dUTP nick end labeling) assay (Supplementary Figs 15 and 16).

**Abscopal effect**. We next examined whether chemotherapy and PDT of NCP@pyrolipid could be used to potentiate a checkpoint blockade therapy such as anti-PD-L1 (α-PD-L1), to enhance the anticancer efficacy and antitumour immunity. A bilateral mouse tumour model of colorectal cancer MC38 was developed by subcutaneously injecting cancer cells into both the left and right flank regions of C57BL/6 mice. The right tumours were designated primary tumours for local light irradiation and the left tumours were designated distant (abscopal) tumours with no direct treatment. When the primary tumours reached ~100 mm³, mice were randomly divided into seven groups ($n=6$): (1) PBS with irradiation, NCP@pyrolipid (2) without or (3) with irradiation, NCP@pyrolipid plus anti-PD-L1 (4) without or (5) with irradiation, (6) porphysome with irradiation plus anti-PD-L1 and (7) oxaliplatin plus porphysome with irradiation plus anti-PD-L1. NCP@pyrolipid, porphysome and oxaliplatin were i.p. injected into animals at an oxaliplatin dose of $2\,mg\,kg^{-1}$ every 3 days for a total of three injections, followed by light irradiation on primary tumours at a light dose of $180\,J\,cm^{-2}$ (670 nm, $100\,mW\,cm^{-2}$) 24 h after injection. After irradiation, mice were immediately i.p injected with PD-L1 antibody at a dose of 50 μg per mouse. Without light irradiation, NCP@pyrolipid alone did not show any inhibition of either primary or distant tumours, as compared with the PBS control group. In contrast, NCP@pyrolipid plus anti-PD-L1 caused significant growth delay in both primary and distant tumours (Fig. 6a,b and

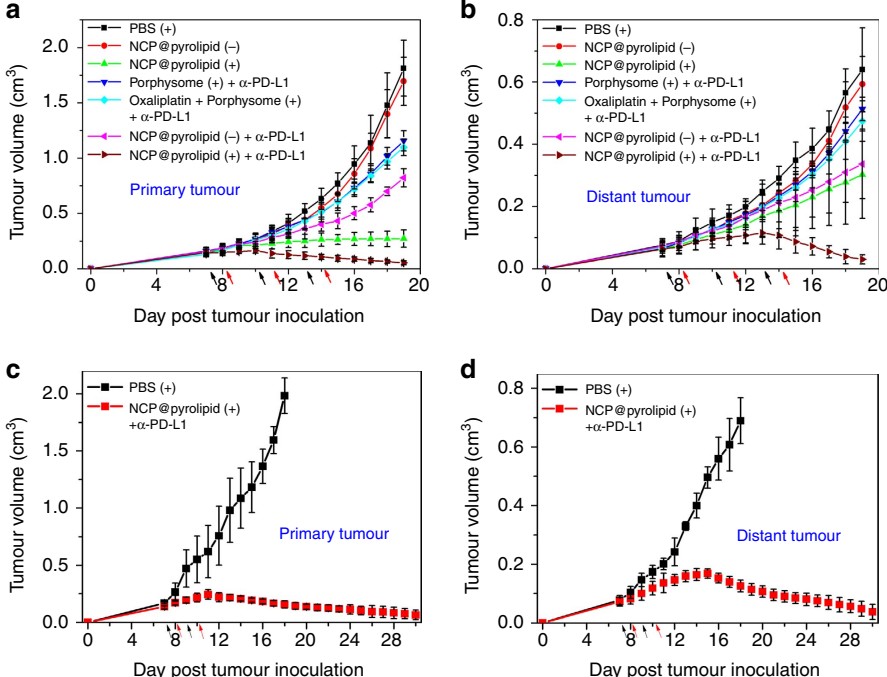

**Figure 6 | The abscopal effect of NCP@pyrolipid in combination with anti-PD-L1.** Bilateral tumour models of MC38 and CT26 were developed by subcutaneously injecting cancer cells into both the right and left flank regions of each animal. The right tumours were designated the primary tumours for light irradiation and the left tumours were designated the distant tumours and did not receive light irradiation. For the MC38 model, PBS, porphysome, oxaliplatin plus porphysome or NCP@pyrolipid was i.p. injected into mice, followed by light irradiation at a dose of 180 J cm$^{-2}$ (670 nm, 100 mW cm$^{-2}$) and i.p. injection of anti-PD-L1 at a dose of 50 µg per mouse. The treatment was carried out every 3 days for a total of three treatments. For the CT26 model, PBS or NCP@pyrolipid was i.p. injected into the mice, followed by light irradiation at a dose of 180 J cm$^{-2}$ (670 nm, 100 mW cm$^{-2}$) and i.p. injection of anti-PD-L1 at a dose of 75 µg per mouse. The treatment was carried out every other day for a total of two treatments. Tumour growth inhibition curves in MC38 (**a,b**) and CT26 (**c,d**) models. The arrows represent the times of drug administration (black) and irradiation (red). '+' and '−' in the figure legends refer to with and without irradiation, respectively. Data are expressed as means ± s.d. ($n = 6$).

Supplementary Figs 17 and 18), indicating that the combination of oxaliplatin with anti-PD-L1 could successfully elicit antitumour immunity. With irradiation but without the checkpoint inhibitor, NCP@pyrolipid was highly effective in inhibiting primary tumour growth and showed a similar effect on the control of the distant tumour growth to treatment with NCP@pyrolipid without irradiation plus anti-PD-L1. This result suggested that oxaliplatin combined with PDT was also able to enhance antitumour immunity. In comparison, NCP@pyrolipid plus anti-PD-L1 with local light irradiation led to efficient tumour regression of the primary tumour with tumours only 2.9% the size of PBS-treated tumours at the endpoint. More importantly, the distant tumours, which did not receive local light irradiation, started to shrink on Day 14 post tumour inoculation and had nearly been eliminated by the endpoint (Day 19). Taken together, the PD-L1 blockade synergized with the abscopal tumour-specific immune response caused by NCP@pyrolipid to mediate regression of both irradiated primary tumours and non-irradiated distant tumours. We also carried out additional control experiments to show that NCP must deliver both oxaliplatin and pyrolipid for effective treatment against tumours: the abscopal effect of oxaliplatin plus porphysome with irradiation plus anti-PD-L1 was similar to that of porphysome with irradiation plus anti-PD-L1 and both produced inferior results to that of NCP@pyrolipid with irradiation plus anti-PD-L1 (Fig. 6a,b). The drastic difference we observed was likely due to the fact that free oxaliplatin and porphysome do not effectively accumulate in tumour tissues.

We then studied the abscopal effect enabled by the combination of NCP@pyrolipid with PDT and anti-PD-L1 on another bilateral syngeneic mouse model of colorectal cancer

CT26. When the primary tumours reached ∼100 mm$^3$, mice received i.p. injections of NCP@pyrolipid at an oxaliplatin dose of 2 mg kg$^{-1}$ every other day, for a total of two injections. Twenty-four hours after injection, the primary tumours were irradiated at a light dose of 180 J cm$^{-2}$ (670 nm, 100 mW cm$^{-2}$). After irradiation, mice were immediately i.p. injected with anti-PD-L1 at a dose of 75 µg per mouse. The combination therapy again led to regression of not only the primary tumours but also the distant tumours after just two treatments (Fig. 6c,d and Supplementary Fig. 19).

**Antitumour immunity of combined NCP@pyrolipid PDT and PD-L1 blockade.** As chemotherapy and PDT of NCP@pyrolipid in combination with anti-PD-L1 caused effective regression of both primary tumours and distant tumours on syngeneic MC38 and CT26 mouse models, which we hypothesized was due to effective systemic antitumour immune responses, we investigated the antitumour immunity induced by chemotherapy/PDT of NCP@pyrolipid in combination with anti-PD-L1 in a syngeneic MC38 mouse model by enzyme-linked immunospot (ELISPOT) and flow cytometry. We first performed an ELISPOT assay to detect the presence of tumour antigen-specific T cells after treatment. Splenocytes were harvested from MC38 tumour-bearing mice treated as described in Fig. 6a,b on Day 19 (12 days after the first treatment) and stimulated with KSPWFTTL, the tumour associated antigen peptide presented by major histocompatibility complex class I (H-2K$^{b}$) for 48 h to detect antigen-specific CD8$^+$ T cells. The number of antigen-specific IFN-γ producing T cells was significantly increased in

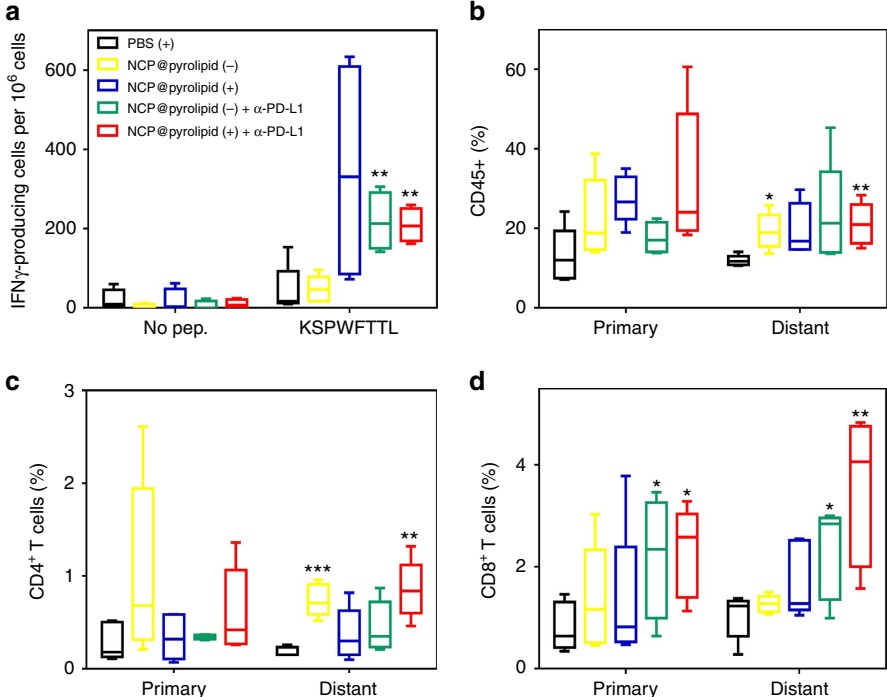

**Figure 7 | Tumour-specific immune responses and the abscopal effect.** Bilateral tumour models of MC38 were established and treated as described in Figs 6a,b. On Day 19 (12 days after the first treatment), the splenocytes were harvested and stimulated with $10\,\mu g\,ml^{-1}$ KSPWFTTL peptide for 48 h. ELISPOT assay was performed to detect IFN-$\gamma$ producing T cells ($n = 4$ or 5) (**a**). The primary (right) and distant (left) tumours were collected for flow cytometry analysis ($n = 5$). The cells were stained with $CD45^+ PI^-$ (**b**), $CD45^+ CD3e^+ CD4^+ PI^-$ (**c**) and $CD45^+ CD3e^+ CD8^+ PI^-$ (**d**), and gated from total tumour cells. Data are expressed as means $\pm$ s.d. ($n = 5$). *$P < 0.05$ from control, **$P < 0.01$ from control and ***$P < 0.001$ from control by $t$-test.

tumour-bearing mice treated with NCP@pyrolipid plus anti-PD-L1 with or without irradiation but not NCP@pyrolipid with or without irradiation (Fig. 7a and Supplementary Fig. 20a). This indicated that NCP@pyrolipid in combination with anti-PD-L1 elicited ICD and effectively generated tumour-specific T-cell response with or without irradiation. We performed flow cytometry analysis to further evaluate the antitumour immune response by elucidating tumour-infiltrating leukocyte profiles. As shown in Fig. 7d and Supplementary Fig. 20d, NCP@pyrolipid in combination with anti-PD-L1 significantly increased the proportion of infiltrating $CD8^+$ T cells in relation to the total number of cells in primary tumours ($t$-test, $P = 4.7 \times 10^{-2}$ without irradiation; $P = 1.1 \times 10^{-2}$ with irradiation) and distant tumours ($t$-test, $P = 2.2 \times 10^{-2}$ without irradiation; $P = 6.3 \times 10^{-3}$ with irradiation), an essential step to induce the abscopal effect. In distant tumours, the percentage of infiltrating $CD45^+$ leukocytes (Fig. 7b and Supplementary Fig. 20b) and $CD4^+$ T cells (Fig. 7c and Supplementary Fig. 20c) with respect to the total number of cells in the tumours were significantly increased in mice treated by PDT of NCP@pyrolipid with anti-PD-L1. Interestingly, PDT of NCP@pyrolipid increased tumour-infiltrating $CD45^+$ leukocytes and $CD4^+$ T cells in distant tumours but had no significant effect on the percentage of $CD8^+$ T cells. No significant difference was observed across the different treatment groups in the amount of tumour-infiltrating B cells (Supplementary Fig. 21) or $CD4^+$ and $CD8^+$ T cells in the lymph nodes (Supplementary Figs 22 and 23).

**Immunofluorescence assay.** The antitumour immune response elicited by chemotherapy and PDT of NCP@pyrolipid in combination with anti-PD-L1 was further confirmed by immunofluorescence assay. We found that NCP@pyrolipid with

irradiation plus anti-PD-L1 treatment instigated TCR $\beta^+$ cell infiltration within both primary and distant tumour tissues, whereas no tumour-infiltrating TCR $\beta^+$ cells were observed in PBS-treated mice. In addition, some of the tumour-infiltrating TCR $\beta^+$ cells were $CD8^+$ (Fig. 8a and Supplementary Fig. 24), indicating the ability of NCP@pyrolipid with irradiation plus anti-PD-L1 to promote $CD8^+$ T-cell infiltration into tumours. The densities of $CD8^+$ T cells in primary and distant tumours treated with NCP@pyrolipid with irradiation plus anti-PD-L1 were calculated to be $168 \pm 33$ and $232 \pm 53$ mm$^{-2}$, respectively (Fig. 8b), which were significantly higher than those in tumours treated with PBS ($15 \pm 12$ mm$^{-2}$ for primary tumours and $30 \pm 15$ mm$^{-2}$ for distant tumours). These results demonstrate that infiltrating $CD8^+$ T cells were significantly increased in mice treated by PDT of NCP@pyrolipid with anti-PD-L1, compared with mice treated with PBS.

## Discussion
As a significant percentage of patients with colorectal cancer die from the metastatic form of the disease[37], it is critical to develop effective treatments that not only eradicate primary tumours but also control metastatic tumours. Our NCP-enabled regimen combines three treatment modalities—chemotherapy by oxaliplatin, PDT by pyrolipid and checkpoint blockade therapy with anti-PD-L1—to achieve superior anticancer efficacy in two syngeneic mouse models of colorectal cancer. NCP@pyrolipid can simultaneously kill cancer cells by inducing apoptosis and stimulate the immune system to activate both acute innate and prolonged adaptive immune responses via synergistic oxaliplatin chemotherapy and pyrolipid-based PDT. PDT of NCP@pyrolipid not only serves as an effective local therapy to eradicate/suppress primary tumour growth but also evokes systemic antitumour

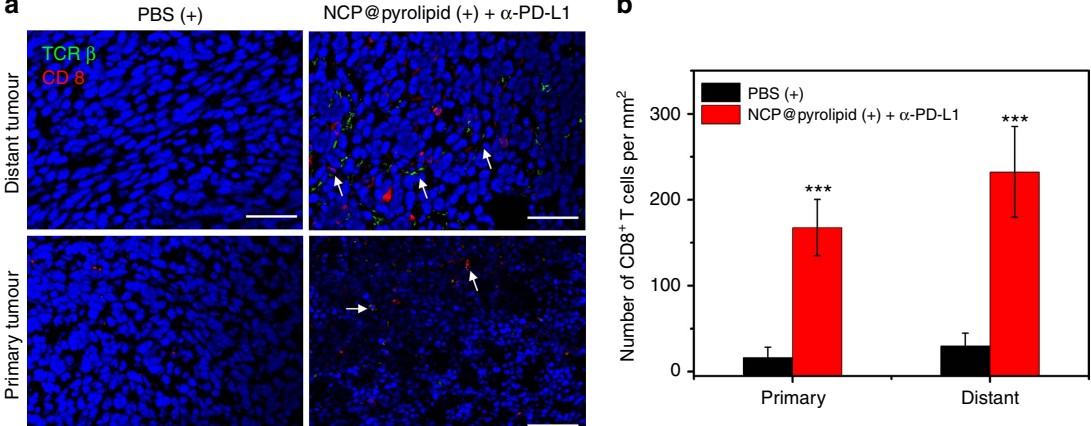

**Figure 8 | CD8$^+$ T cells immunofluorescence assay.** Bilateral tumour models of MC38 were established and treated with PBS with irradiation or NCP@pyrolipid with irradiation plus anti-PD-L1. On Day 19 (12 days after the first treatment), primary (right) and distant (left) tumours were collected, sectioned and subjected to immunofluorescence staining. (**a**) Representative CLSM images of tumours after immunofluorescence staining. White arrows indicate CD8$^+$ T cells. Tumour cell nuclei in treated primary tumours appear to be smaller, probably due to the effects of PDT treatment. Scale bar, 50 µm. (**b**) The densities of CD8$^+$ T cells in the whole tumours. Data are expressed as means ± s.d. ($n = 3$). ***$P < 0.001$ from control by $t$-test.

immunity, which further potentiates PD-L1 checkpoint blockade therapy.

Our understanding of cellular and molecular tumour immunology has evolved dramatically over the past two decades, which has enabled the identification of new and innovative ways to manipulate the immune response to cancer.[38–43] Most immunotherapies target the immune system but not the cancer and, therefore, immunotherapies are believed to be a promising foundation to build treatment regimens for a variety of tumour types[44–47]. Clinical results suggest that immunotherapies have potential for durable and adaptable cancer control at different stages of the disease[48]. To maximize benefits, however, combination regimens with conventional cancer treatments that operate by distinct mechanisms will be necessary, to increase the durable response rate of cancer immunotherapies[7].

NCP@pyrolipid nanoparticles self-assembled into core-shell structures with an asymmetric lipid bilayer coating that carried 27.6 wt% oxaliplatin in the core and released its cargo intracellularly. By passive targeting via the enhanced permeability and retention effect, NCP@pyrolipid achieved significantly higher cellular uptake of oxaliplatin and pyrolipid than other nanoparticle or free drug formulations. The efflux of oxaliplatin and pyrolipid was negligible, probably due to the partial incorporation of lipids into cell membranes during internalization that may have modified the membrane structure and prevented oxaliplatin or pyrolipid from effluxing out of the cells[18]. With optimal particle size, surface properties and stability, NCP@pyrolipid exhibited long blood circulation half-lives for both oxaliplatin and pyrolipid to leverage passive targeting, resulting in high tumour uptake of 10.4 ± 0.7% ID g$^{-1}$ 24 h with low MPS clearance after i.p. injection.

Oxaliplatin is a Food and Drug Administration-approved chemotherapeutic drug for the treatment of colorectal cancer, known to induce cell death by triggering apoptosis and to stimulate pre-apoptotic CRT exposure, a distinct marker for ICD[26,35]. The exposure of CRT on the cell surface serves as an 'eat me' signal to dendritic cells and macrophages[29]. Mature dendritic cells migrate to the lymph node, where they prime naive T cells into effector T cells, which migrate to the tumour microenvironment[34]. NCP@pyrolipid leverages the immune system during chemotherapy by converting apoptotic death from 'silent' to immunogenic, thus acting as an 'anticancer vaccine.' The CRT exposure demonstrated by flow cytometry and CLSM, and successful prevention against tumour challenge by PDT of NCP@pyrolipid proved the effective ICD induced by the treatment.

As a local therapy, pyrolipid-enabled PDT[33] also contributes to enhanced antitumour immunity by three mechanisms. First, PDT exerts systemic influence by promoting secretion of chemokines and cytokines, which stimulates the immune system to exert antitumour activity[36]. We observed significantly elevated pro-inflammatory cytokines TNF-α, INF-γ and IL-6 one day after PDT treatment, followed by a rapid drop in cytokine levels 2 days after PDT treatment (Fig. 3b–d), suggesting that the treatment evoked acute inflammation to prompt an innate immune response. Second, PDT has been found to induce ICD and thus activate the immune system[36]. Third, PDT of NCP@pyrolipid kills cancer cells by both apoptosis and necrosis (Supplementary Table 1). The innate immune effector cells engulf portions of the stressed and dying necrotic tumour cells and present tumour-derived antigenic peptides to T cells, thus stimulating a tumour-specific T-cell response[36].

PDT of NCP@pyrolipid in combination with anti-PD-L1 treatment presents three regimens—oxaliplatin, PDT and checkpoint blockade therapy—to elicit synergistic effects in enhancing antitumour immunity for the effective treatment of metastatic colorectal cancer. PD-1 is a cell-surface co-inhibitory receptor expressed on T cells, B cells, monocytes and natural killer cells, and it has two known ligands, PD-L1 and PD-L2. PD-L1 is upregulated by tumour cells and by cells in the tumour microenvironment[49]. Multiple preclinical studies demonstrated that blockade of the interaction between PD-1 and PD-L1 using anti-PD-1 or anti-PD-L1 can restore T-cell activity against tumour cells, thereby preventing cancer metastasis and reducing tumour volume[50,51]. Infiltrating T cells and PD-L1 expression are essential for PD-L1 blockade therapy to be effective but are only found in immunogenic tumour microenvironment[15,52,53]. As a result, checkpoint blockade cancer therapy is only effective in patients whose tumours are immunogenic, which might explain the low rate of durable responses in clinical trials. We hypothesize that the response rate and efficacy of PD-L1 checkpoint blockade therapy can be improved when used in combination with therapies designed to create an immunogenic tumour microenvironment, eventually leading to durable clinical

benefits. We believe that chemotherapy/PDT of NCP@pyrolipid provides an efficient way to induce immunogenicity in the tumour microenvironment and enhance antitumour immunity of anti-PD-L1 to empower checkpoint blockade cancer therapy.

We have elucidated the general principle that our NCP@pyrolipid can enhance the efficacy of PD-L1 checkpoint blockade therapy. The number of antigen-specific IFN-γ producing T cells and CD8$^+$ T cells were significantly increased in tumour-bearing mice treated with NCP@pyrolipid with irradiation plus anti-PD-L1, as shown by ELISPOT assay, flow cytometry assay and inmmunofluorescence staining (Figs 7 and 8). Galon and coworkers[54–57] have elegantly shown that the type, density and location of immune cells within human colorectal tumours are a better predictor of patient survival than the histopathological methods currently used to stage colorectal cancer. We thus intend to examine the changes in the tumour environment in our future work instead of the broader immune responses, although we also recognize that the immunological data in humans are considerably different to a subcutaneous colon cancer mouse model.

In summary, we have developed an effective NCP-enabled combination therapy for metastatic colorectal cancer that combined oxaliplatin chemotherapy, pyrolipid-based PDT and PD-L1 checkpoint blockade cancer therapy. NCP@pyrolipid carried high amounts of oxaliplatin and pyrolipid that showed prolonged blood circulation and favourable tumour accumulation after systemic administration. PDT of NCP@pyrolipid effectively inhibited tumour growth in subcutaneous CT26 and HT29 mouse models. More importantly, both oxaliplatin and PDT contributed to an immunogenic environment in the tumour, which significantly enhanced PD-L1 checkpoint blockade therapy by generating systemic antitumour immunity. As a result, PDT of NCP@pyrolipid in combination with anti-PD-L1 regressed the growth of not only primary tumours but also distant tumours in two bilateral syngeneic mouse models of colorectal cancer. We believe the combination of chemotherapy, PDT and checkpoint blockade therapy designed in the current study offer a new strategy for treating many metastatic cancers with primary tumours accessible by PDT.

## Methods

**Cell lines and animals.** Human colorectal adenocarcinoma cell HT29 and murine colon adenocarcinoma cell CT26 and MC38 cells were obtained from the American Type Culture Collection (Rockville, MD), tested for mycoplasma contamination and cultured in McCoy's 5A, RPMI 1640 and DMEM medium, respectively, supplemented with 10% fetal bovine serum (FBS).

Athymic male nude mice (6 weeks, 24–26 g), BALB/c male mice (6 weeks, 20–22 g) and C57BL/6 mice (6 weeks, 20–22 g) were provided by Harlan-Envigo Laboratories, Inc. (USA). The study protocol was reviewed and approved by the Institutional Animal Care and Use Committee at the University of Chicago.

**Preparation and characterization of NCP@pyrolipid.** DOPA-capped NCP nanoparticles were prepared according to our previous report[16]. NCP@pyrolipid was prepared by adding a THF solution (80 μl) of 1,2-distearoyl-sn-glycero-3-phosphocholine, cholesterol, pyrolipid, 1,2-distearoyl-sn-glycero-3-phosphoethanolamine polyethylene glycol 2000 (2:1:0.8:1 in molar ratio) and DOPA-coated NCP to 500 μl of 30% (v/v) ethanol/water at 60 °C. The mixture was stirred at 1,700 r.p.m. for 1 min. THF and ethanol were completely evaporated and the NCP@pyrolipid solution was allowed to cool to room temperature (r.t.). NCP@pyrolipid was centrifuged at 19,650 g. for 30 min. The supernatant was then removed and the particles resuspended in PBS. ICP-MS (Agilent 7700X, Agilent Technologies, USA) was used to analyse the Pt concentration of NCP to calculate oxaliplatin loading. The particle size and zeta potential of NCP@pyrolipid in PBS were determined by Zetasizer (Nano ZS, Malvern, UK). Transmission electron microscopy (Tecnai Spirit, FEI, USA) was used to observe the morphology of NCP@pyrolipid.

**Cytotoxicity of NCP@pyrolipid in colorectal cancer cells.** The cytotoxicity of NCP@pyrolipid was tested in CT26 and HT29 cells. The cells were seeded on 96-well plates at 2,500 cells per well. After incubating for 24 h, the cells were treated with NCP@pyrolipid, porphysome, NCP and free oxaliplatin at various oxaliplatin concentrations or pyrolipid concentrations. After a 24 h incubation, the cells were irradiated with LED light (670 nm) at 60 mW cm$^{-2}$ for 15 min (equals to 54 J cm$^{-2}$). The cells without irradiation treatment served as controls. The cells were further incubated for 48 h. The cell viability was detected by MTS (3-(4,5-dimethylthiazol-2-yl)-5-(3-carboxymethoxyphenyl)-2-(4-sulfophenyl)-2H-tetrazolium) assay (Promega, USA) and the IC$_{50}$ values were calculated accordingly.

**In vitro ICD.** The ICD induced by NCP@pyrolipid was evaluated by immunofluorescence and flow cytometry. For immunofluorescence analysis, CT26 cells were seeded at $5 \times 10^5$ cells per well in six-well plates and further cultured for 24 h. The culture media were replaced by 2 ml of fresh culture media containing 10% FBS. Oxaliplatin, NCP, NCP@pyrolipid or porphysome was added to the cells at an equivalent oxaliplatin dose of 5 μM and pyrolipid dose of 1.6 μM. Cells incubated with PBS served as a control. After a 24 h incubation, the cells were irradiated with LED light (670 nm) at 100 mW cm$^{-2}$ for 15 min (equal to 90 J cm$^{-2}$). Following a further incubation of 4 h, the cells were washed with PBS three times, fixed with 4% paraformaldehyde, incubated with Alexa Fluor 488-CRT antibody for 2 h, stained with DAPI and observed under CLSM using 405 and 488 nm lasers for visualizing nuclei and CRT expression on the cell membrane, respectively. For flow cytometry analysis, CT26 cells were seeded at $1 \times 10^6$ cells per well in six-well plates and further cultured for 24 h. The culture media were replaced by 2 ml of fresh culture media containing 10% FBS. Oxaliplatin, NCP, NCP@pyrolipid and porphysome were added to the cells, respectively, at an equivalent oxaliplatin dose of 5 μM and pyrolipid dose of 1.6 μM. Cells incubated with PBS served as a control. After a 24 h incubation, the cells were irradiated with LED light (670 nm) at 100 mWcm$^{-2}$ for 15 min (equal to 90 J cm$^{-2}$). Following a further incubation of 4 h, the cells were collected, incubated with AlexaFluor 488-CRT antibody for 2 h and stained with PI. The samples were analysed by flow cytometry (LSRII Orange, BD, USA), to identify cell surface CRT. The fluorescence intensity of stained cells was gated on PI-negative cells.

**Pharmacokinetics and biodistributions.** Mice were subcutaneously injected in the right flank with 1 million CT26 cells and tumours were allowed to grow to ~100 mm$^3$ before they received i.v. administration of NCP@pyrolipid at an oxaliplatin dose of 3 mg kg$^{-1}$. Animals were killed (three per time point) at 5 min, 1, 3, 8, 24 and 48 h after nanoparticle administration. After collecting the blood, the livers, lungs, spleens, kidneys, bladders and tumors were harvested. The organs, tumors and blood were digested in concentrated nitric acid for 24 h and the Pt concentrations were analysed by ICP-MS. The pyrolipid amounts in the blood collected at 5 min, 1, 3, 8, 24 and 48 h were determined using the same extraction and detection method as the recovery experiment, as we previously reported[18]. Briefly, the blood was centrifuged at 4,535 g. for 10 min to separate plasma. Methanol and 0.25% Triton X-100 was added to the plasma to extract the pyrolipid and prevent aggregation. Pyrolipid concentrations were determined by ultraviolet–visible spectroscopy.

**In vivo anticancer efficacy.** The PDT efficacy of NCP@pyrolipid was investigated using the HT29 subcutaneous xenograft mouse model and CT26 flank tumour syngeneic mouse model. Tumours were established in mice by subcutaneous inoculation with HT29 cell suspension ($2 \times 10^6$ cells per mouse) or CT26 cell suspension ($1 \times 10^6$ cells per mouse) into the right flank region of 6-week athymic male nude mice or 6-week BALB/c male mice, respectively. Four groups were included for comparison: PBS with irradiation as a control; NCP with irradiation; NCP@pyrolipid with irradiation; and NCP@pyrolipid without irradiation. When tumours reached 100 mm$^3$, NCP or NCP@pyrolipid was i.v. injected into animals at an oxaliplatin dose of 2 mg kg$^{-1}$ every 4 days for a total of two injections for the CT26 tumour model and a total of four injections for the HT29 tumour model. Twenty-four hours after injection, mice were anaesthetized with 2% (v/v) isoflurane and tumours were irradiated with a 670 nm LED for 30 min. The energy irradiance was measured to be 100 mW cm$^{-2}$ and the total light dose was 180 J cm$^{-2}$.

To evaluate the therapeutic efficacy, tumour growth and body weight evolution were monitored. Tumour size was measured with a digital caliper every day. Tumour volumes were calculated as follows: (width$^2$ × length)/2. All mice were killed when the tumour size of the control group exceeded 2 cm$^3$ and the excised tumours were photographed and weighed. The tumours were embedded in optimal cutting temperature medium, sectioned at 5 μm thickness and subjected to haematoxylin and eosin stain for histopathological analysis and TUNEL (Invitrogen, USA) assay for quantifying the in vivo apoptosis. Livers, lungs, spleens and kidneys were also excised after the mice were killed and then embedded in optimal cutting temperature medium, sectioned at 5 μm thickness, stained with haematoxylin and eosin and observed for toxicity with light microscopy (Pannoramic Scan Whole Slide Scanner, Perkin Elmer, USA). For the CT26 mouse model, blood was collected on Days 7, 8, 9 and 10, and the serum TNF-α, IFN-γ and IL-6 production was determined by enzyme-linked immunosorbent assay (R&D Systems, USA), to evaluate the immunogenic response evoked by the treatment.

**Abscopal effect on MC38 model.** C57BL/6 mice were injected subcutaneously with $5 \times 10^5$ MC38 cells into the right flank (primary tumour) and $1 \times 10^5$ MC38 cells into the left flank (secondary tumour). When the primary tumours reached $\sim 100\,mm^3$, mice were randomly divided into seven groups ($n = 6$): PBS with irradiation as control; NCP@pyrolipid without irradiation; NCP@pyrolipid with irradiation; NCP@pyrolipid without irradiation plus anti-PD-L1; NCP@pyrolipid with irradiation plus anti-PD-L1; porphysome with irradiation plus anti-PD-L1; and oxaliplatin plus porphysome with irradiation plus anti-PD-L1. NCP@pyrolipid, porphysome and oxaliplatin were i.p. injected into animals at an oxaliplatin dose of $2\,mg\,kg^{-1}$ every 3 days for a total of three injections. Twenty-four hours after injection, mice were anaesthetized with 2% (v/v) isoflurane and primary tumours were irradiated with a 670 nm LED at a light dose of $180\,J\,cm^{-2}$ given at $100\,mW\,cm^{-2}$. After irradiation, mice were immediately i.p. injected with PD-L1 antibody at a dose of $50\,\mu g$ per mouse. Primary and secondary tumour sizes and mouse body weights were monitored every day. Tumour size was measured with a digital caliper and calculated as follows: $(width^2 \times length)/2$. All mice were killed when the primary tumour size of the control group exceeded $2\,cm^3$.

**Abscopal effect on CT26 model.** BALB/c mice were injected subcutaneously with $1 \times 10^6$ CT26 cells into the right flank (primary tumour) and $2 \times 10^5$ CT26 cells into the left flank (secondary tumour). When the primary tumours reached $\sim 100\,mm^3$, mice were i.p. injected with NCP@pyrolipid at an oxaliplatin dose of $2\,mg\,kg^{-1}$ every 2 days for a total of two injections. Twenty-four hours after injection, mice were anaesthetized with 2% (v/v) isoflurane and primary tumours were irradiated with a 670 nm LED at a light dose of $180\,J\,cm^{-2}$ given at $100\,mW\,cm^{-2}$. After irradiation, mice were i.p. injected immediately with PD-L1 antibody at a dose of $75\,\mu g$ per mouse. The primary and secondary tumour sizes and mouse body weights were monitored every day. The tumour size was measured with a digital caliper and calculated as follows: $(width^2 \times length)/2$. All mice were killed when the primary tumour size of the control group exceeded $2\,cm^3$.

**ELISPOT assay.** Tumour-specific immune responses to IFN-γ were measured *in vitro* by ELISPOT assay (Mouse IFN gamma ELISPOT Ready-SET-Go!; Cat. No. 88-7384-88; eBioscience). A Millipore Multiscreen HTS-IP plate was coated overnight at 4 °C with anti-Mouse IFN-γ capture antibody. Single-cell suspensions of splenocytes were obtained from MC38 tumour-carrying mice and seeded onto the antibody-coated plate at a concentration of $2 \times 10^5$ cells per well. Cells were incubated with or without KSPWFTTL stimulation ($10\,\mu g\,ml^{-1}$; in purity >95%; PEPTIDE 2.0) for 48 h at 37 °C and then discarded. The plate was then incubated with biotin-conjugated anti-IFN-γ detection antibody at r.t. for 2 h, followed by incubation with Avidin-HRP for 2 h at r.t. 3-amino-9-ethylcarbazole substrate solution (Sigma, Cat. AEC101) was added for cytokine spot detection.

**Flow cytometry.** Tumour-draining lymph nodes were harvested and ground with the rubber end of a syringe. Tumours were harvested, treated with $1\,mg\,ml^{-1}$ collagenase I (Gibco, USA) for 1 h and ground with the rubber end of a syringe. Cells were filtered through nylon mesh filters and washed with PBS. The single-cell suspension was incubated with anti-CD16/32 (clone 93; eBiosciences) to reduce nonspecific binding to FcRs. Cells were further stained with the following fluorochrome-conjugated antibodies: CD45 (30-F11), CD3e (145-2C11), CD4 (GK1.5), CD8 (53-6.7), B220 (RA3-6B2) and PI staining solution (all from eBioscience). LSR FORTESSA (BD Biosciences) was used for cell acquisition and data analysis was carried out using FlowJo software (TreeStar, Ashland, OR).

**Immunofluorescence assay.** Tumours were collected and frozen tissue sections of $6\,\mu m$ thickness were prepared using a cryostat. These sections were air-dried for at least 1 h and then fixed in acetone for 10 min at $-20$ °C. After blocking with 20% donkey serum, the sections were incubated with individual primary antibodies against TCRβ (eBioscience) and CD8 (Thermo Scientific) overnight at 4 °C, followed by incubation with dye-conjugated secondary antibodies for 1 h at r.t. After staining with DAPI for another 10 min, the sections were then washed twice with PBS and observed under CLSM (Olympus, FV1000).

**Data availability.** The authors declare that all the data supporting the findings of this study are available within the article and its Supplementary Information files or from the corresponding author upon reasonable request.

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

## Acknowledgements

We thank Dr Haidong Tang for experimental help and acknowledge the National Cancer Institute (U01–CA198989), the University of Chicago Medicine Comprehensive Cancer Center (NIH CCSG: P30 CA014599), the Cancer Research Foundation and the Ludwig Institute for Metastasis Research for funding support.

## Author contributions

C.H., X.D. and N.G. contributed equally. W.L. and C.H. conceived the project. C.H., X.D., N.G., C.C. and C.P. performed the experiments and analysed the results. C.H., X.D., N.G., C.C., R.R.W. and W.L. wrote the manuscript.
