## [Peer Review File · Nature Communications]

Reviewers' comments:

Reviewer #1 (Expert in Nanoparticles/PDT and cancer) (Remarks to the Author):

Summary

This communication is a tour-de-force in cancer nanomedicine/therapy. It combines 3 modes of therapy, Chemo, photodynamic and immune therapy, in an original and smart way, which, moreover turns out to be quite promising, i.e., effective. Specifically, the authors prepared PDT/Chemo-NP. It is composed of Zn core Phospholipids NP. Pyrolipid (Porphyrin conjugated with Phospholipid) was used for PDT and Oxaliplatin was used for Chemotherapy. In addition to PDT/Chemo, the authors introduced PD-L1 antibody therapy (inducing programmed cell death) and showed impressive tumor reduction in vivo (figure 6).

Comments

λ Whenever there is a combination of 3 factors, in principle there is the need to compare with any of the combinations of just 2 factors, as well as with single factors. While a good job was done on the single factors, a clearer discussion on the relative effectiveness of each of the pairwise combinations would help, and is called for. From the Pharmacokinetics results, the NP does not seem to have specific affinity toward the tumor, so the question of potential side effects is relevant. What are the systemic effects from the oxaliplatin? Or do the authors claim that the NP is acting as a tumor targeted vehicle?

λ Authors need to discuss more re the design of the nanoparticles. The authors added Triton X-100 to the matrix to disrupt the micelle formed. However, how is this relevant to the in vivo settings?

λ Authors need be more specific re what pyrolipid is and what it is composed of.

λ Authors need to discuss more regarding the release of the oxaliplatin from the nanoparticle. How is oxaliplatin released from the system and where in the NP is it located (core or shell)?

λ Please define "Nanoscale coordination polymer (NCP) core-shell nanoparticles."

Reviewer #2 (Cancer Immunotherapy)(Remarks to the Author):

The authors developed a combination treatment regimen for metastatic colorectal cancer consisting of NCP@pyrolipid and PDL1. The authors convincingly demonstrated in mouse models that NCP@pyrolipid with oxaliplatin induces immunogenic cell death at the tumor site. It is believed that this mechanisms leads tu cross priming, enhanced anti tumor T cell responses that can be enhanced by PDI1.

The clinical situation in humans that suffer from metastatic CRC is more complex. We know from Galon and halama that T cells are detectable at the tumor site, we learned that T cell densities represent the strongest prognostic factor and T cekk density at the invasive margin of metastatic lesions is the strongest predictor for response to chemotherapy.

So there is obviously good immunity in metastatic disease and the challenge is not so much to induce broader responses, the challenge is to change the environment and make that more accessible for effector T cells.

The treatment developed by the authors might help there, but unfortunately they did not focus on analyzing the specific changes in the tumorenvironment induced by the treatment.

I would like to know

1. How do the numbers/densities of T cells, and all other immune cell subsets change under treatment (density, spatial distribution)
2. What is the exact effect on the cytokine/chemokine profile in the tumor (separately center, invasive margin, adjacent normal).

I am not convinced that the therapeutic effect is mainly triggered by induction of broader immunity and if this is the case I fear this is not relevant for the human situation

Reviewers' comments:

Reviewer #1 (Expert in Nanoparticles/PDT and cancer) (Remarks to the Author):

Summary

This communication is a tour-de-force in cancer nanomedicine/therapy. It combines 3 modes of therapy, Chemo, photodynamic and immune therapy, in an original and smart way, which, moreover turns out to be quite promising, i.e., effective. Specifically, the authors prepared PDT/Chemo-NP. It is composed of Zn core Phospholipids NP. Pyrolipid (Porphyrin conjugated with Phospholipid) was used for PDT and Oxaliplatin was used for Chemotherapy. In addition to PDT/Chemo, the authors introduced PD-L1 antibody therapy (inducing programmed cell death) and showed impressive tumor reduction *in vivo* (figure 6).

Comments

Whenever there is a combination of 3 factors, in principle there is the need to compare with any of the combinations of just 2 factors, as well as with single factors. While a good job was done on the single factors, a clearer discussion on the relative effectiveness of each of the pairwise combinations would help, and is called for. From the Pharmacokinetics results, the NP does not seem to have specific affinity toward the tumor, so the question of potential side effects is relevant. What are the systemic effects from the oxaliplatin? Or do the authors claim that the NP is acting as a tumor targeted vehicle?

Response: We thank the reviewer for strong endorsement of our work and insightful suggestions. We have added the groups porphysome(+)+ α -PD-L1 and oxaliplatin+porphysome(+)+ α -PD-L1 to further illustrate the effectiveness of our nanoparticle formulation in enhancing anticancer efficacy and antitumor immunity, with additional discussion in the revised manuscript. For the 2-factor combinations, NCP@pyrolipid(+) is the combination of chemotherapy and PDT while NCP@pyrolipid(-)+ α -PD-L1 is the combination of chemotherapy and immunotherapy. The added porphysome(+)+ α -PD-L1 group is the last two-factor combination of PDT and immunotherapy.

The accumulation of NCP@pyrolipid is based on the EPR effect. NCP@pyrolipid is stable due to the lipid layer and PEG coating, as shown from the drug release study, which enabled long circulation time (half-life 11.8 ± 1.9 for oxaliplatin) and low uptake by the MPS as evidenced by the low accumulation in liver, spleen, and kidney. So NCP@pyrolipid is acting as a passive, but nonetheless strong targeting vehicle; however, we did not emphasize this point in our manuscript, because many NPs have some passive targeting capacity. Due to the high tumor-to-normal tissue ratio and the local application of light, the NCP@pyrolipid is very safe for *in vivo* application, as shown by the consistent body weight (Figure S18 and S19).

Authors need to discuss more re the design of the nanoparticles. The authors added Triton X-100 to the matrix to disrupt the micelle formed. However, how is this relevant to the *in vivo* settings?

Response: *In vitro*, we studied the difference in the properties of particles with intact or disrupted lipid layer, so we added Triton X-100 to disrupt the lipid layer. *In vivo*, we do not use Triton X-100. The lipid layer on the NCP surface can improve the stability and prolong the circulation time, thereby enhancing

the tumor accumulation due to the EPR effect. When NCP@pyrolipid entered cancer cells via endocytosis *in vivo*, the lipid layer might dissociate and incorporate into the cell and plasma membranes, making the particle more easily internalized by cancer cells and then release oxaliplatin into the cancer cells. No Triton X-100 is needed to disrupt the lipid layer *in vivo*.

Authors need be more specific re what pyrolipid is and what it is composed of.

Response: Pyrolipid is a pyropheophorbide-lipid conjugate, which is generated by an esterification reaction between 1-palmitoyl-2-hydroxy-sn-glycero-3-phosphocholine (P-lysoPC) and pyropheophorbide, a chlorophyll-derived porphyrin analogue. We added the information to our revised manuscript.

Authors need to discuss more regarding the release of the oxaliplatin from the nanoparticle. How is oxaliplatin released from the system and where in the NP is it located (core or shell)?

Response: As we said in the manuscript, NCP@pyrolipid is a core-shell nanostructure with NCP carrying oxaliplatin as its solid core and a self-assembled asymmetric lipid bilayer incorporating pyrolipid as its shell. The NCP cores were constructed from the coordination between Zn^{2+} and phosphate groups of the oxaliplatin prodrug, so oxaliplatin is located in the core of NCP.

We also ran the release profiles of oxaliplatin from DOPA-NCP (without the outer lipid layer) and NCP@pyrolipid in PBS. DOPA-NCP revealed rapid burst release, with 76% cumulative release of oxaliplatin within 2 h. In contrast, the release of oxaliplatin for NCP@pyrolipid is slow and sustained, with only 5.6% and 21% oxaliplatin released within 2 h and 96 h, respectively. The remarkable decrease in the release of drugs from lipid coated particles was most likely due to the lipid layer located at the surface, which could not only increase the stability and prevent particles dissociation but also forms an effective physical barrier to impede drug diffusion. We added the method, results and discussion in the revised manuscript and supporting information.

Please define "Nanoscale coordination polymer (NCP) core-shell nanoparticles."

Response: "Nanoscale coordination polymer (NCP) core-shell nanoparticles" is defined as nanoparticles composed of a coordination polymer (repeating units of Zn ions linked to the phosphate groups of the oxaliplatin prodrug) in its core and a lipid bilayer as a shell.

Reviewer #2 (Cancer Immunotherapy) (Remarks to the Author):

The authors developed a combination treatment regimen for metastatic colorectal cancer consisting of NCP@pyrolipid and PDL1. The authors convincingly demonstrated in mouse models that NCP@pyrolipid with oxaliplatin induces immunogenic cell death at the tumor site. It is believed that this mechanisms leads tu cross priming, enhanced anti tumor T cell responses that can be enhanced by PDL1.

The clinical situation in humans that suffer from metastatic CRC is more complex. We know from Galon and halama that T cells are detectable at the tumor site, we learned that T cell densities represent the strongest prognostic factor and T cekk density at the invasive margin of metastatic lesions is the strongest predictor for response to chemotherapy.

So there is obviously good immunity in metastatic disease and the challenge is not so much to induce broader responses, the challenge is to change the environment and make that more accessible for effector T cells.

Response: We observed significant CD4⁺ T cell infiltration into both primary and distant tumors, suggesting that our treatment indeed altered the tumor microenvironment to make them more accessible for effector T cells. Please see the next response for details.

The treatment developed by the authors might help there, but unfortunately they did not focus on analyzing the specific changes in the tumor environment induced by the treatment.

I would like to know

1. How do the numbers/densities of T cells, and all other immune cell subsets change under treatment (density, spatial distribution)

Response: We determined the density as % of tumor cells and % of lymph nodes of CD4⁺ and CD8⁺ T cells under various treatment conditions by flow cytometry, as shown in Figures 7 and S20-22. We qualitatively supported our flow cytometry results by looking at immunofluorescence (Figure 8) staining for CD3e and CD8 in the treatment group with the highest response (NCP@pyrolipid(+)+ α -PD-L1) compared to PBS treated control to evaluate both the density and spatial distribution of the immune cell subsets within the tumor. Immunofluorescence of the tumor slices show CD8⁺ T cells in the center of the primary treated tumor, whereas the CD8⁺ population is fairly dispersed in the distant untreated tumor and can be visualized both in the center of the tumor and at the edges. Based on work by Galon et al. (Science. 2006; 313(5795):1960-4.) showing that tumors from patients without recurrence had higher immune cell densities, we believe that our observed increase in immune cell densities (2.2% compared to 0.81% for control and 1.39% compared 0% for control by flow cytometry and CLSM, respectively) suggests our combination therapy is a promising treatment method. Our flow cytometry results (Figure 7) also show an increase in infiltrating CD4⁺ T cells in the primary and, significantly, in the distant tumors by combination therapy.

2. What is the exact effect on the cytokine/chemokine profile in the tumor (separately center, invasive margin, adjacent normal).

Response: We recognize the importance in understanding the cytokine/chemokine profile and spatial distribution in the tumor, given the differences in cytokine expression between microsatellite instable and microsatellite stable colorectal cancer in the human condition (Oncoimmunology. 2014;3:e29256. eCollection 2014). Our work focuses on treating subcutaneous tumors, which we expect to be distinct from human tumor biopsies or orthotopic murine models. We appreciate the suggestion and hope we can delve deeper into understanding the immunological response in future works.

I am not convinced that the therapeutic effect is mainly triggered by induction of broader immunity and if this is the case I fear this is not relevant for the human situation

Response: We respect the reviewer's reservations about potential pitfalls in the event of translating this combination therapy to human beings. With a subcutaneous colon cancer mouse model, we were primarily concerned with elucidating general principles of immune activation by the combination of nanoparticle-mediated chemotherapy and PDT and how it interacts with checkpoint blockade therapy. We look forward to studying details of the tumor microenvironment now that we have established we can trigger a broad immune response with our therapy.

REVIEWERS' COMMENTS:

Reviewer #1 (Remarks to the Author):

Overall, the paper has now been improved, meeting my previous objections. The exception is the "indirect targeting" of the nanoparticles by the EPR effect. This may make a big difference regarding the efficacy of these nanoparticles. The authors invoke it twice in their defense given in their "rebuttal", but surprisingly they seem not to have included any of it in the revised manuscript. I recommend to include it explicitly in the paper.

Reviewer #2 (Remarks to the Author):

The authors have addressed the critical issues raised by the reviewers properly. The more detailed analysis of the effects on the immune cell composition adds significant more details that are convincing that there is a therapeutic effect.
I would accept the manuscript

REVIEWERS' COMMENTS:

Reviewer #1 (Remarks to the Author):

Overall, the paper has now been improved, meeting my previous objections. The exception is the "indirect targeting" of the nanoparticles by the EPR effect. This may make a big difference regarding the efficacy of these nanoparticles. The authors invoke it twice in their defense given in their "rebuttal", but surprisingly they seem not to have included any of it in the revised manuscript. I recommend to include it explicitly in the paper.

Response: We have now added two statements in the discussion (in the third paragraph) to point out passive targeting of our particles. The second sentence starts with "By passive targeting via the enhanced permeability and retention (EPR) effect," and the fourth sentence ends with "...to leverage passive targeting, resulting high tumor uptake..."

Reviewer #2 (Remarks to the Author):

The authors have addressed the critical issues raised by the reviewers properly. The more detailed analysis of the effects on the immune cell composition adds significant more details that are convincing that there is a therapeutic effect.

I would accept the manuscript

Response: We thank the reviewer for positive endorsement of our work.